# Study on the Purity of Gold Leaf in a SO_2_ Atmosphere at Ambient Temperature

**DOI:** 10.3390/ma14092425

**Published:** 2021-05-06

**Authors:** Houyi Huang, Guanglin Xu, Xinyou Liu

**Affiliations:** 1College of Art and Design, Nanjing Forestry University, Nanjing 210037, China; huanghouyi@njfu.edu.cn; 2Co-Innovation Center of Efficient Processing and Utilization of Forest Resources, Nanjing Forestry Univesity, Nanjing 210037, China; xuguanglin@njfu.edu.cn; 3Academy of Chinese Ecological Progress and Forestry Studies, Nanjing Forestry University, Nanjing 210037, China; 4College of Furnishing and Industrial Design, Nanjing Forestry University, Str. Longpan No.159, Nanjing 210037, China

**Keywords:** gold leaf, SO_2_ atmosphere, corrosion, humidity

## Abstract

Gold leaf samples of different purities were corroded in a SO_2_ atmosphere at three different relative humidities (30%, 60%, 90%) at ambient temperature, and the effects on color, gloss, and morphology were studied. Results showed that a corrosion rate of 0.0898 g/cm^2^ could be attained after 6 weeks at high humidity. Color changes also occurred during the gold leaf corrosion process, and many thin pits formed on the surfaces, as shown by SEM. EDX results showed that these pits contained C, O, and S compounds. By comparing the results of different gold purity samples and different relative humidity conditions, it could be concluded that both gold content and humidity play an important role in SO_2_ atmosphere corrosion. These conclusions are helpful for the conservation of gold leaf decorative cultural relics.

## 1. Introduction

Gold is an important material from ancient times and has been found as a decorative material in tombs from the Shang Dynasty (1600BC–1046BC) [1,2,3]. Gold is widely used in various decorations, such as Chinese classical furniture [4,5], due to its brilliant color, dazzling luster, and excellent chemical properties. Gold leaf gilding, which is the art of decorating the whole or parts of wood, metal, plaster, glass, or other objects with gold in either leaf or powder form [6,7,8,9], is an important gold decoration process. Gold leaf techniques imitating the appearance of solid gold have been used in all cultures that have placed a high value on gold.

The traditional process of making gold leaf is based on using high-purity gold with 99.99% gold content as the main raw material. Through a special procedure involving 12 processes (see Figure 1), such as rationing gold, casting, rolling, hammering, and leaf cutting, gold leaf presents a bright golden color, soft texture, low density, and is less than 0.12 μm in thickness [10,11,12]. As the main component of gold leaf, gold has the advantages of having stable properties and a resistance to color change, oxidation, moisture, corrosion, mildew, insect bites, and radiation [13].

Theoretically, gold can only react with one kind of acid, namely aqua regia. However, the electrolytic corrosion of gold products in an acidic gas environment has attracted extensive attention [14,15,16]. It seems that gold can only be corroded in a single acid at room temperature. According to relevant reports, many cultural relics decorated with gold leaf were damaged to varying degrees when exposed to air. For example, the gold leaf-decorated Dazu thousand-hand Buddha statue in Chongqing, China, which was listed as a world cultural heritage site in 1999, has been seriously damaged in recent years due to the acidic climate [17,18,19]. The statue of Buddha has been in the cave for more than a thousand years. The accelerated damage in recent years suggests that the damage may be related to recently changing environmental conditions. According to environmental studies, the annual average relative humidity is 82%, and it can reach 90% in both winter and spring. In recent years, more than 70% of the rainwater in this area has been shown to be acidic, and the pH value of the rainwater is 4.16–4.46. Acid fog, caused by acid rain, covering the surface of the gold leaf for a long period of time, may be the main cause of gold leaf damage [20,21]. SO_2_ is the precursor of environmental acidification and the direct product of stone fuel combustion. With the rapid development of the modern industry, the SO_2_ content in the air has gradually increased. It is easy to form acid in a humid environment that corrodes environmental facilities, especially metal products [22,23,24].

In fact, in the process of making gold leaf, the physical and chemical properties of gold have changed. According to the traditional Chinese process, gold leaf must be heated to 800 °C for softening, hammering, cooling, and annealing. The gold undergoes rapid deformation many times, and, ultimately, the thickness of the gold leaf changes from 80 m to 100 nm. During reheating, cooling, and hammering, the high-density crystallinity of gold atoms and subfeatures such as vacancies, dislocations, grain boundaries, and subgrain boundaries, are destroyed [10]. To show different colors on the surface of the gold leaf and improve the hardness of the gold leaf, silver, copper, platinum, and other metal elements are often added into the gold leaf, which will reduce its purity and lead to decreased corrosion resistance. In addition, other impurities in gold leaf will also reduce its corrosion resistance [25,26,27,28].

In view of the above situation, the content and relative humidity of acid gas in the atmosphere and gold purity in the gold leaf are important factors affecting the corrosion resistance of gold leaf. This paper aims to explore the corrosion behavior of gold leaf of different purities in a SO_2_ environment under ambient temperature and different levels of humidity, which will deepen the understanding of the nature of gold leaf and contribute to the conservation of gold leaf decoration relics. At the same time, it can also provide useful guidance for the conservation of modern gold leaf decorative art.

## 2. Materials and Methods

### 2.1. Materials

The gold leaf was provided by Nanjing General Plant of Gold Leaf and Wire, Jiangsu, China, with the codes 99#, 91#, and 74# (according to the producer, the gold contents of these gold leaf samples are 99%, 91%, and 74%). They were prepared with a traditional fabrication technique at a thickness of 95 nm. A sulfur dioxide atmosphere was prepared with analytical reagents, Na_2_SO_3_ (purity ≥97%) and 65% H_2_SO_4_, provided by Tianjin Kemiou Chemical Reagent Co., Ltd., Tianjin, China. SO_2_ gas could be produced by the following reaction (1):Na_2_SO_3_ + H_2_SO_4_ = Na_2_SO_4_ + H_2_O+ SO_2_↑(1)

### 2.2. Corrosion Test

The gold leaf simulation corrosion test device was 2 L, as shown in Figure 2. The gold leaf was wound around a rectangular sheet of acrylonitrile butadiene styrene (ABS) plastic (2 mm thickness) with dimensions of 35 mm × 60 mm. These samples were placed vertically on an ABS support with grooves where the samples could be fixed. The relative humidity in the SO_2_ atmosphere was controlled by a petri dish containing an aqueous glycerine solution according to ASTM D5032-1997 (2003) [29]. The selected relative humidities were 30 ± 5%, 60 ± 5%, and 90 ± 5%. Ten grams of Na_2_SO_3_ was carefully added to the bottom of the kettle, and 15 mL of analytical reagent H_2_SO_4_ was carefully dropped into the kettle to react with Na_2_SO_3_. When the reagent acids were added, both the reacted and resultant gas were observed. Then, the kettle cap was applied and screwed, and the resultant product gas could be observed, being released through the valve on the kettle cap. The valve was then closed when no gas release was observed, and the concentration of sulfur dioxide was approximately 5.0 g/L. The samples were removed after six weeks for characterization. 

### 2.3. Corrosion Rate

The corrosion rate (CR) was determined based on weight before and after corrosion. The CR was calculated by Equation (2):CR = (w_1_ − w_0_)/A(2)
where w_0_ is the weight of the gold leaf and the ABS plastic sheet before corrosion testing (g) with a precision of 0.0001 g, w_1_ denotes the weight of specimens after corrosion (g) with a precision of 0.0001 g, and A is the surface of the gold leaf (mm^2^) with a precision of 0.0001 mm^2^.

### 2.4. Color Measurement

Color measurements of all specimens (20 pieces for each case) were taken on the surface before and after corrosion testing using an AvaSpec-USB2 spectrometer, equipped with an integrating AVA sphere with a diameter of 80 mm, interconnected by optical fibers. Measurements were made using a D65 illuminant and a 10° standard observer. The percentage of reflectance, collected at 10 nm intervals over the visible spectrum (from 400 to 700 nm), was converted into the CIELAB color system. These color coordinates are: lightness L (varying from 0 for black to 100 for white), redness a (varying from negative values for green to positive values for red on the green–red axis), and yellowness b (varying from negative values for blue to positive values for yellow on the blue-yellow axis). All samples were measured both initially and after different humidity corrosion testing. Color differences between treated and control samples were calculated based on Equation (3):∆E = (∆L^2^ + ∆a^2^ + ∆b^2^)^1/2^(3)
where ∆L, ∆a, and ∆b are the differences in the initial and final values (before and after treatment) of the L, a, and b parameters, respectively. A low ∆E value corresponds to a low color difference.

### 2.5. Gloss Evaluation

The evaluation of surface gloss was performed with a gloss meter Micro-Tri-Gloss No. 4520 (BYK Additives & Instruments, Geretsried, Germany). After equipment calibration, all samples were measured with a square measurement area of 2 mm × 4 mm, at 20°, 60°, and 85° incidence angles. Then, a comparative study was performed on the treatment method and t-test at *p* = 0.05 in Statistical Analysis System (SAS) (v. 9.4, SAS Institute, Cary, NC, USA).

### 2.6. Morphological Characteristics and XPS Analysis

The surface shapes of the samples were assessed based on morphology using a cold field emission scanning electron microscope (FE-SEM, HITACHI Regulus 8100, Tokyo, Japan) to investigate potential variations in the physical structures. The electron beam acceleration was set at 0.1 kV to 30 kV. In addition, a working distance of approximately 1.5–30 mm, a resolution of 1.0 nm, and a minimum calibration period of 10 nm were applied to the experiments. To evaluate the microstructure and surface composition of the samples before and after corrosion testing, an Oxford Instruments Ultim Max 170 energy-dispersive X-ray spectroscopy (EDX) detector was attached to the FE-SEM. X-ray photoelectron spectroscopy (XPS) spectra were recorded on a Kratos Axis Ultra DLD spectrometer (Kratos Analytical Ltd., Chestnut Ridge, NY, USA), and the binding energy was calibrated by O 1 s and Au 4 f peaks.

## 3. Results and Discussion

### 3.1. Corrosion Rate

Corrosion rate shows the mass change in gold leaf per unit area, which is an important index to evaluate the corrosion degree of gold leaf. A greater mass change implies a higher corrosion degree; on the contrary, a lower mass change means a lower corrosion degree. Figure 3 shows that all gold leaf samples suffered corrosion under the SO_2_ atmosphere. The average corrosion rates of the 74# gold leaf samples were 0.0445 g/cm^2^, 0.0512 g/cm^2^, and 0.0756 g/cm^2^ under various relative humidities (30%, 60%, and 90%, respectively) in SO_2_, while the corrosion rates of the 99# gold leaf samples were 0.0205 g/cm^2^, 0.0378 g/cm^2^, and 0.0544 g/cm^2^ under 30%, 60%, and 90% relative humidity, respectively. In a period of six weeks, the corrosion degree of the 91# gold leaf samples was the highest, with a corrosion rate of 0.0898 g/cm^2^ under 90% relative humidity. It is likely that impurities in the 91# gold leaf absorb the most hydrogen, oxygen, and sulfur elements during corrosion at 90% humidity. Compared with the values of corrosion rate under a 30% relative humidity SO_2_ atmosphere, corrosion rates of samples under a 60% and 90% relative humidity SO_2_ atmosphere were much higher. The corrosion rate values of the 99# gold leaf samples were lower than those of 91# and 74# at each humidity level, which may be correlated with the gold purity. Therefore, it is evident that a SO_2_ atmosphere causes gold leaf corrosion, where gold purity and relative humidity are important factors affecting the degree of corrosion.

### 3.2. Color Measurement

The main component of gold leaf is gold, which is similar in color to its namesake. To improve the hardness of gold leaf, silver, copper, platinum, and other metals are typically added, so the color of the gold leaf will also change accordingly. The L, a, and b coordinates of the 99# gold leaf samples before corrosion testing were 57.23, 13.87, and 34.28, respectively, while the corresponding values for the 91# and 74# gold leaf samples were 58.79, 10.28, and 35.44, and 64.32, 2.69, and 20.85, respectively. It was observed that when the L values increased, the values of a and b decreased, as the gold ratio increased in the gold leaf.

Figure 4 shows the change in L, a, b, and total color change (ΔE) of the samples after corrosion testing. The ΔL resulted in positive values after corrosion testing, indicating a lighter color of the gold leaf following corrosion via decreased color intensity during and after the corrosion process. On the contrary, the color coordinates a and b decreased by −3.00 units for Δa and −2.51 units for Δb for the 99# gold leaf under a 90% relative humidity SO_2_ atmosphere. Compared with corrosion testing under a 30% relative humidity SO_2_ atmosphere, the color changes were larger under a 90% relative humidity SO_2_ atmosphere with a maximum 5.26 unit change for 99# gold leaf.

### 3.3. Gloss Evaluation

Gloss is used to characterize the specular ability of the surface to reflect light. High gloss is an important attribute of gold leaf. Table 1 shows the gloss of the specimens before and after corrosion testing under various incidence angles of 20°, 60°, and 85°. The average gloss values of the 99# gold leaf samples before corrosion testing were 982.5, 715.5, and 135.5 under incidence angles of 20°, 60°, and 85°, respectively.

At the same respective incidence angles, the gloss values of the 91# and 74# gold leaf samples were 897.8, 712.7, and 135.9, and 986.0, 746.1, and 132.8, respectively. After six weeks under a SO_2_ atmosphere, all samples lost glossiness to different degrees. At a 20° incidence angle, 99# gold leaf lost glossiness in the range 5.40–20.64%, 91# gold leaf lost glossiness in the range of 2.64–19.26%, and 74# gold leaf lost glossiness in the range of 7.71–10.04%. At incidence angles of 60° and 85°, the glossiness changes showed the same trend. Comparing the relative humidity conditions, the gloss of the gold leaf increased with increasing relative humidity. Therefore, corrosion markedly decreased the gloss of gold leaf.

### 3.4. Morphology

The SEM micrographs of the samples corroded in the SO_2_ atmosphere and those of the control group are illustrated in Figure 5. Although the gold leaf is largely even, many wrinkles, tiny pits, and rough surfaces of the gold leaf can be observed in the micrographs of the control groups. Therefore, it can be deduced that many crystal defects stem from the large amount of cold deformation during the production process. The results demonstrate that the corrosion tests had a remarkable influence on the tissue structure of the samples. In the micrographs of corroded groups, many pits (marked with circles) can be found through SEM observation. These pits might be related to a loss of gloss.

### 3.5. EDX and XPS Analysis

Energy-dispersive X-ray spectroscopy is an important method used to evaluate the microstructure and surface composition of samples before and after corrosion testing. EDX surface scanning of three randomly selected regions of the control and corroded gold leaf was conducted. 

The average elemental compositions of the control and corroded gold leaf in selected regions are noted in Table 2. Compared with the control groups, the surfaces of corroded gold showed increasing O, C, and S content. Therefore, it can be assumed that the corrosion products are O, C, and S-containing compounds.

Figure 6 shows the XPS spectra calibrated by O 1 s and Au 4 f peaks of the control and corroded samples under 90% relative humidity. The proportion of oxygen ions in all gold leaf samples of different purities increased, which illustrates that an oxidation reaction occurred during the corrosion test. Moreover, after the corrosion test, the relative mass of the gold decreased. The mass of the gold should be constant, which also explains the increase in the weight of the gold leaf during corrosion. Comparing the different gold purities, the relative mass of gold in 74# decreased the most.

## 4. Conclusions

SO_2_ plays an important role in the corrosion of gold leaf. After 6 weeks in a SO_2_ environment inducing corrosion, the mass change in gold leaf can reach 0.0898 g per cm^2^. The colors of the gold leaf also changed after corrosion, and the basic change trend was that L values increased and a and b values decreased. The decrease in gloss indicated a change in the surface of the gold leaf, where the products of the corrosion process affected the specular reflection of light on the surface of the gold leaf. SEM images showed that the surface of the gold leaf formed many pits after corrosion, which also explained the change in gloss. EDX results showed that the pits produced by corrosion contained C, O, and S compounds. The results of the gold leaf corrosion experiments using different gold purities under different relative humidities were compared. Higher gold purities and a lower relative humidity led to lower degrees of corrosion. On the contrary, lower gold purities and a higher relative humidity led to higher degrees of corrosion. The relative mass of oxygen increased, as found by EDX and XPS, indicating that oxidation occurs during the corrosion process. These conclusions should be helpful for the conservation of gold leaf decorative cultural relics. 

## Figures and Tables

**Figure 1 materials-14-02425-f001:**
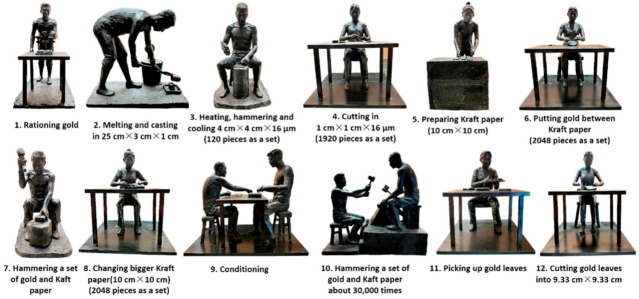
Schematic diagram of Chinese traditional gold leaf forging process.

**Figure 2 materials-14-02425-f002:**
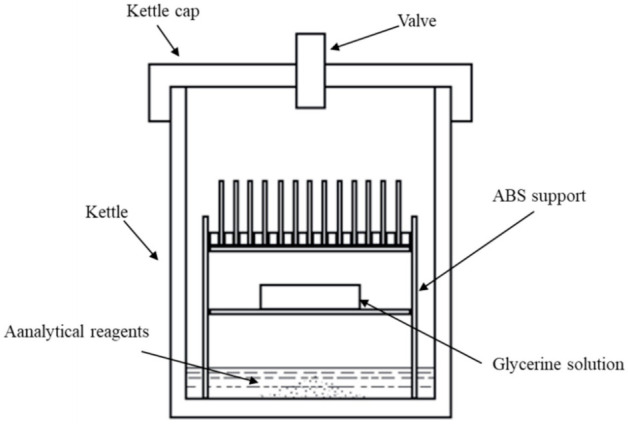
Schematic diagram of the simulated corrosion test device.

**Figure 3 materials-14-02425-f003:**
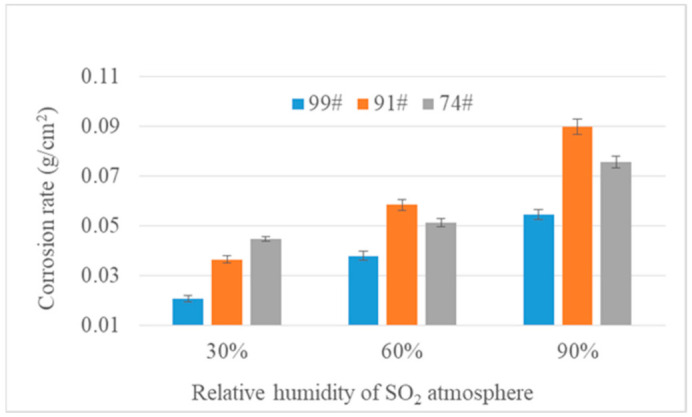
Corrosion rate of various gold leaf samples in a SO_2_ atmosphere over six weeks.

**Figure 4 materials-14-02425-f004:**
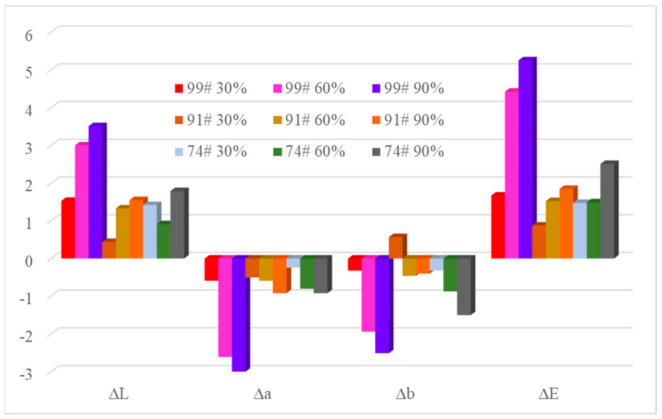
Change in L, a, and b and total color change (ΔE) in samples after corrosion testing.

**Figure 5 materials-14-02425-f005:**
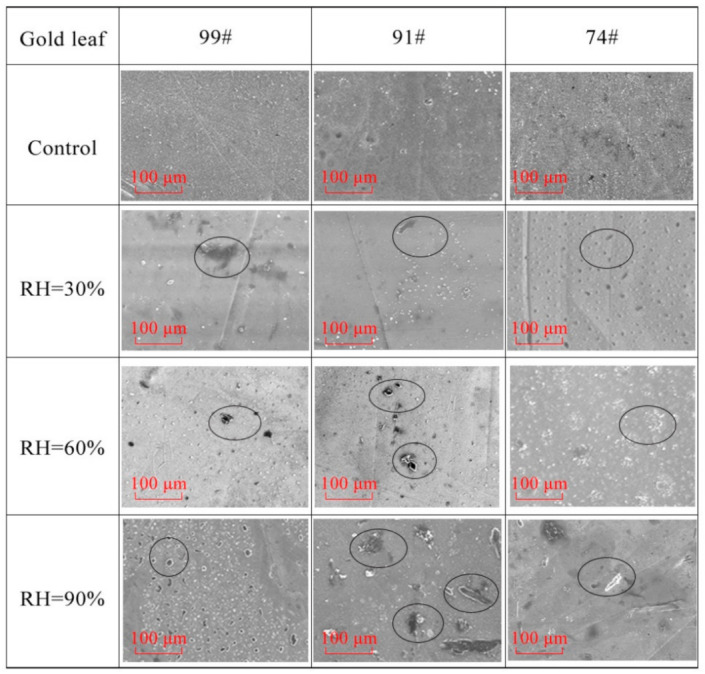
SEM micrographs of the control group and corroded group (magnification = 1000×).

**Figure 6 materials-14-02425-f006:**
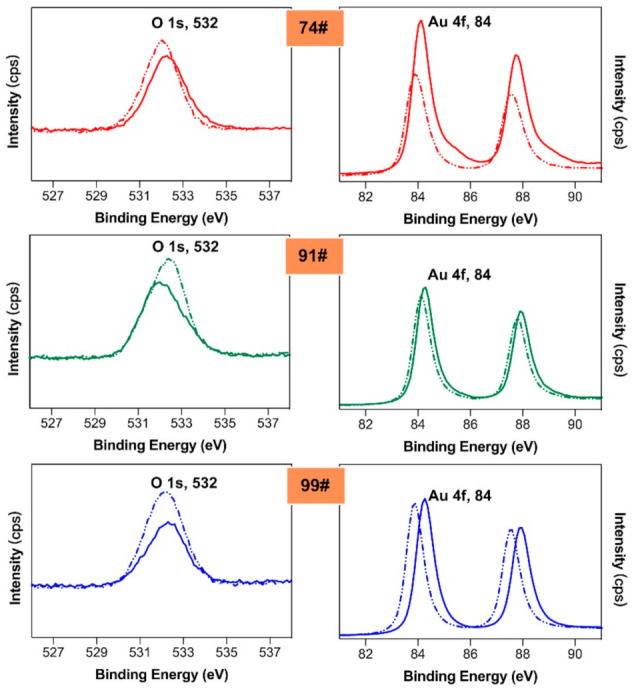
XPS spectra at O 1 s and Au 4 f regions of control (solid lines) and corroded (dash lines) sample under 90% relative humidity.

**Table 1 materials-14-02425-t001:** Gloss of uncorroded and corroded samples (mean ± standard deviation).

Incidence Angles →		20°	60°	85°
Gold Leaf ↓	Condition ↓			
99#	Control	982.5 ± 8.1	715.5 ± 7.8	135.5 ± 3.7
	30%	929.4 ± 7.6	696.0 ± 6.4	126.8 ± 2.9
	60%	856.5 ± 8.3	634.8 ± 5.9	124.5 ± 3.2
	90%	779.7 ± 6.7	592.5 ± 5.6	121.7 ± 2.3
91#	Control	897.8 ± 5.9	712.7 ± 6.3	135.9 ± 4.2
	30%	874.2 ± 5.3	629.8 ± 5.2	132.6 ± 2.7
	60%	779.3 ± 6.1	627.8 ± 6.1	129.4 ± 2.9
	90%	724.9 ± 4.7	623.8 ± 6.6	127.4 ± 1.9
74#	Control	986.0 ± 9.6	746.1 ± 6.4	132.8 ± 5.1
	30%	910.0 ± 10.2	720.2 ± 5.9	129.8 ± 3.1
	60%	896.1 ± 8.7	711.9 ± 5.3	128.1 ± 3.4
	90%	887.0 ± 9.3	703.3 ± 6.7	127.3 ± 2.1

**Table 2 materials-14-02425-t002:** Average elemental composition of the control and corroded samples in selected regions.

Chemical Elements →		Au	Pt	Ag	Cu	O	C	S
Gold Leaf ↓	Condition ↓							
99#	Control	95.47	1.91	1.23	1.34	0.02	0.03	0.00
	30%	88.12	1.74	1.14	1.30	5.94	1.23	0.53
	60%	86.88	1.73	1.11	1.24	6.04	2.11	0.89
	90%	85.20	1.69	1.04	1.31	7.22	2.31	1.23
91#	Control	86.45	2.33	5.70	4.93	0.12	0.47	0.00
	30%	81.33	2.19	5.34	5.22	3.98	0.99	0.95
	60%	79.24	2.14	5.22	5.06	4.73	2.34	1.27
	90%	77.03	2.08	5.03	4.96	6.21	3.15	1.54
74#	Control	68.73	4.26	20.31	5.33	0.78	0.59	0.00
	30%	63.86	3.96	19.87	5.23	4.56	1.68	0.84
	60%	60.43	3.75	17.86	5.89	8.47	2.53	1.07
	90%	56.95	3.52	16.84	5.59	11.56	3.85	1.69

## Data Availability

Data is contained within the article.

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
