# Peer review of "Study on the Purity of Gold Leaf in a SO2 Atmosphere at Ambient Temperature"

_materials, 2021, doi:10.3390/ma14092425_

Round 1
Reviewer 1 Report
The presented to Materials journal manuscript 1187923 aims to explore the corrosion behavior of gold leaves of different purities in a SO2 environment under ambient temperature and different humidity.
I would recommend this manuscript to be published in Materials journal after major corrections.
In this article it is not proved that there is a corrosion of gold, because it is not shown the evidence of the variation of its oxidation state. The increase in mass and the variation in color and brightness can simply be the result of the deposition of species on the leaf surface, or even chemical reactions on its surface without gold intervening as a reagent. You will have to do XPS or another technique to prove it.
Another important thing. In general, it is considered that gold cannot be attacked except by aqua regia. Even the Pourbaix diagram shows this. But the scenario is different when gold complexes can be formed. The classical Pourbaix diagram only considers the presence of water and O2. In the presence of conditions where gold complexes can be formed, the standard potential is very different and oxidation becomes possible. Authors should look for this information and include it in the article. This is very important and should be discussed in the manuscript.
Photos, i.e. SEM images (line 201) are also of poor quality. Labels or description for three columns of images are missing. Do they correspond to the gold specimens with different purities? Letter “R” is missing before “H=90%”.
It would be good to show optical images of the gold after the attack and before the attack.
References. The list has different fonts used for typing. Please, use the same one.
Finally, the idea of the presented manuscript is similar to that of Ref [10] Gold Leaf Corrosion in Moisture Acid Atmosphere at Ambient Temperature, Rare Metal Materials and Engineering, 43 (2014) 2637-2642. When modifying your work, try to focus on new insights about gold corrosion and not repeat it.
Good work!
Author Response
Thank you for your helpful review. According to your helpful comments and suggestions, the XPS analysis was added and SEM images were also modified.
Reviewer 2 Report
- The objects of research are the gold leaves. It is necessary to present the appearance of the product and indicate how it is made, since mechanical processing also affects the rate of corrosion.
- It remains unclear what was the concentration of SO2?
- Figure 2 shows that a product made of 91 % gold corrodes faster at 60 and 90% humidity. An explanation or suggestion should be given as to why this may be caused.
- In Figure 3, the histograms of ΔL* for 99 % gold at 60 and 90% humidity are colored pink and purple. You should also do the same for other parameters, since orange and gray are already used.
- In the Conclusions it would be necessary to make a forecast of the service life of the products in the test environment.
Author Response
Thank you for your helpful review. According to your helpful comments and suggestions, the manuscript was modified as follow.
- Chinese traditional gold leaf forging process was added.
- The concentration of SO2 was added.
- An explanation was given as to the product made of 91 % gold corrodes faster at 60 and 90% humidity. why this may be caused.
- The histograms of ΔL* for 99 % gold at 60 and 90% humidity were changed as the same color.
- The conclusions was also revised.
Round 2
Reviewer 1 Report
Thank you for your efforts in improving the manuscript.
However, the quality of the images was not changed. On the pdf document that I have access, the scheme (Fig.1), labels on bar graphs, on SEM images and lines on XPS spectra (no differences between dash and solid lines are detectable) are poorly visible. These images should be inserted into the final document with better quality, otherwise it will be difficult for readers to look at them. I insist the authors provide better images.
Author Response
Thank you very much for your comments. Fig.1, SEM images and XPS spectra all were changed.
Reviewer 2 Report
The explanation of why gold leaves with a gold content of 91 % corrode faster at 60 and 90% humidity is insufficient. Perhaps this can be a topic for further research. Predicting the durability of products, that is, determining the period when they will need to be restored or replaced, can also become a topic of further research.
Author Response
Thank you very much for you suggestion. Predicting the durability of products decorated with gold leaves will be our future research.